# Convolution Neural Network Development for Identifying Damage in Vibrating Pylons with Mass Attachments

**DOI:** 10.3390/s24196255

**Published:** 2024-09-27

**Authors:** George D. Manolis, Georgios I. Dadoulis

**Affiliations:** Laboratory for Experimental Strength of Materials and Structures, School of Civil Engineering, Aristotle University of Thessaloniki, GR-54124 Thessaloniki, Greece

**Keywords:** machine learning, convolution neural networks, autoencoders, principal component analysis, damage detection, structural dynamics, pylons, vibrations, fracture

## Abstract

A convolution neural network (CNN) is developed in this work to detect damage in pylons by measuring their vibratory response. More specifically, damage detection through testing relies on the development of damage-sensitive indicators, which are then used to reach a decision regarding the existence/absence of damage, provided they have been retrieved from at least two distinct structural states. Damage indicators, however, exhibit a relatively low sensitivity regarding the onset of structural damage, further exacerbated by the low amplitude response to a variety of environmentally induced loads. To this end, a mathematical model is developed to interpret the experimental data recovered from a fixed-base pylon with a top mass attachment to transverse motion. Damage is introduced in the mathematical model in the form of springs corresponding to the cracking of the beam’s lower end. Families of numerically generated acceleration records are produced at select stations along the beam’s height, which are then used for training a CNN. Once trained, it is used to identify damage from acceleration records produced from a series of experiments. Difficulties faced by CNN in correctly identifying the presence/absence of damage in the pylon are discussed, and steps taken to improve the quality of the results are proposed.

## 1. Introduction

As infrastructure continues to age, structural health monitoring (SHM) techniques are being developed [1] to observe structural components and entire systems over time, recording their responses. These responses are then compared against established benchmarks to determine if the structure is under stress, which would require a detailed inspection first, followed by necessary repairs [2]. With the vast amount of data collected over time, recent advancements in hardware have made it feasible to use wireless sensors that can transmit signals to a remote processing unit, where the structural integrity is assessed. To manage the extensive data flow over time, sensors can be equipped with software that analyzes the monitored structure. This initial evaluation can filter out routine responses caused by environmental factors, allowing only significant data to be sent to the remote processing unit [3]. Over time, machine learning techniques such as neural networks have been brought into the picture to help evaluate the recorded data. These are being incorporated into existing SHM systems and are further enriched with explainable artificial intelligence (XAI) capabilities [4], which can suggest reasons for the divergence of the structural response from prescribed norms and advise as to why a specific repair protocol should be followed. Finally, structural control mechanisms can be implemented, ranging from passive to active systems, whereby the latter case requires continuous monitoring for its activation by tracking vibrations before they become harmful to the structure’s integrity [5].

When focusing on pylons, recent literature highlights advancements in vibration control for wind turbine towers, which include passive, active, and semi-active methods, with the latter two requiring monitoring of the structural response. These methods are crucial for vibration mitigation, as they address a wide range of loads, including wind, waves, currents, and ground motions [6]. They are also used in a variety of engineering disciplines. In mechanical engineering, for example, machine learning techniques have been applied for real-time condition monitoring of rotating shafts, emphasizing the importance of accurately identifying damage through vibration measurements for predictive maintenance [7]. Continuous monitoring of wind turbines has also shown variability in their natural frequencies and damping ratios, allowing for enhanced tracking of modal properties through operational modal analysis [8]. Automated procedures are nowadays developed to define reference modal properties for modal tracking, facilitating the detection of problematic resonances [9].

Pylons can be viewed as various types of beams, which are essentially unidimensional structural elements that experience tension, compression, bending, and twisting forces. The use of beams is widespread in civil engineering but is also found in mechanical engineering (torsion bars, stringers, etc.) and in aeronautical engineering (turbine blades, struts, etc.). A few parameters enter the design of such structural elements, starting with the type and shape of the cross-section (hollow, non-uniform, etc.), type of construction (uniform, segmented, etc.), type of material (e.g., steel, concrete, composites), their support conditions and the presence of attachments (e.g., secondary systems). Beams subjected to transient loads are of particular interest, as they can be accurately modeled using the elastic waveguide approach [10]. Depending on the specific beam and type of load applied, the three fundamental modes of mechanical behavior may become interrelated. Additionally, the Bernoulli–Euler beam theory may require modifications to account for the rotational inertia of the cross-section (Rayleigh theory) when the load frequencies are high and to incorporate shear deformation (Timoshenko theory) if the beam is not slender. Furthermore, material damping within the beam must be considered, often modeled for waveguides using the Kelvin–Voigt viscoelasticity model, which results in complex-valued material constants [11].

Finally, it is essential to define the impact of lumped mass attachments on pylons. They are primarily equipment serving various purposes, e.g., lights, transmitters, cameras, etc. However, they may also play the role of a passive or semi-active structural control device [12]. A recent approach [13] explores the influence of time-varying mass attachments on flexible pylons undergoing ground-induced vibrations placed within the context of active control. An analytical solution is derived for a lumped mass at the top of a cantilevered pylon experiencing longitudinal vibrations, highlighting the complications in the modal analysis due to time-dependent system matrices. Results indicate the frequency range where a varying mass optimally modifies the pylon’s kinematic response. This methodology can also be extended to include transverse and rotational vibrations.

In addition, the attached mass may, in turn, be connected to a spring and/or to a damper. For instance, ref. [14] examined a nonlinear pendulum with such an absorber attached to its mass. Generally, a flexural beam with an attached mass can be considered a coupled system consisting of a primary linear oscillator and a nonlinear secondary system. Significant research has been conducted in this area, particularly when the secondary system is treated as a nonlinear energy sink [15]. Naturally, new solution methods are required, such as multiple-scale expansions that simplify the problem. The first approximation is a low-dimension vibro-impact system with a force and a damper, which can be solved analytically. However, these analyses are most appropriate when the mass of the attached system is small relative to that of the primary system.

Damage in beams due to the development of Mode I (open/closed) fatigue cracks leads to noticeable changes in the vibration frequencies of the structural system. Specifically, cracks in the web or connections of steel and reinforced concrete (R/C) beams—such as those used in girders, bridge decks, and pylons—significantly affect their vibratory response, weaken the structure, and can eventually lead to local failure [16]. A fundamental mechanical model for simulating cracking in Bernoulli–Euler beams involves introducing discontinuities at the suspected crack locations [17]. There are four primary types of discontinuities caused by cracks, which are modeled by incorporating the Heaviside function into the displacement, slope, bending moment, or shear force diagrams of the beam. The most common approach is modeling slope discontinuities, which introduce Dirac delta function representations in the bending moment and shear force diagrams. An engineering approach to modeling cracks, therefore, involves inserting springs at the crack location, with the type of spring depending on the specific discontinuity expected to occur [18]. Additionally, the Dirac delta function and its first derivative can be treated as forcing functions on the right-hand side of the Bernoulli–Euler equation, maintaining the self-adjoint property of the differential operator [19] during solution.

In this work, we focus on examining the structural integrity of pylons with variable cross-sections that are subjected to environmentally induced vibrations. Four scenarios are investigated, namely (a) the intact pylon; (b) the intact pylon with an attached point mass at the top that might serve a variety of uses, e.g., lighting equipment; (c) damage in the form of a crack at the base when recordings are available in the same direction as the crack itself; and (d) the same case as before but with recordings available in the direction perpendicular to the crack. Obviously, the first two scenarios involve a linear response of the pylon, while the remaining two scenarios address nonlinear mechanical behavior in the pylon, the reason being localized plastic flow in the vicinity of the crack’s tips plus friction across facing crack surfaces as the pylon vibrates. An engineering approximation for modelling these types of problems is using spring elements, while the current trend is to base the overall assessment of the mechanical response exclusively on data-driven methods [20]. More specifically, data-driven methods are used when the governing equations of the problem at hand are not known. From the response to a given input, optimization is carried out by assuming a matrix function and sequentially adjusting its coefficients until the error between what is measured and what is assumed is minimized. The data drive part comes in assuming the form of this matrix function. This restricts the choices to ones that are most probably correct. For instance, in structural dynamics, this matrix function should be symmetric and positive definite. This corresponds to the eigenvalue problem for structural dynamics and makes the entire procedure for developing a CNN more efficient.

Typical data-driven methods rely on a first assessment of the eigenvalue-eigenvector pairs of the structure in question under different conditions. In this notion, any deviations observed between these pairs can be classified in terms of their causal factors. Data-driven methods are capable of extracting eigenvectors even if the mechanical response of the structure in question is nonlinear, essentially by piecewise linearization of the eigenvalue problem. Since our experimental data used as input contains nonlinearities, it was essential to use this type of method. Therefore, eigenvector extraction will be accomplished by the use of *autoencoders* [21]. See Figure 1 for a schematic of their architecture. Every *autoencoder* comprises two mappings, ${\ddot{q}}_{en}:\;\ddot{w}\to \;\Phi$ and ${\ddot{q}}_{de}:\;\Phi \to \hat{\ddot{w}}\;$, where q, Φ, w pertain to the generalized beam coordinates, the eigenfunctions, and the transverse displacements, respectively, with overdots indicating time derivatives. From the architecture shown in Figure 1, the weights of the decoder, that is, the mapping ${\ddot{q}}_{de}$, are the generalized coordinates of the corresponding eigenfunction.

## 2. Industrial Pylon Case Study

Any CNN development pre-supposes a linear *activation* function plus a *cost* function interposed between the *input* and the *reconstruction* segments, see [22]. By modelling the flexible pylon shown in Figure 2 as a continuous cantilevered beam, its transverse acceleration $\ddot{w}$ response can be written in terms of the spatially-dependent eigenfunctions $\Phi_{j}(x)$ and the time-dependent, generalized coordinates $q\left(t\right)$ as $\ddot{w}\left(x,t\right)=\sum_{j=1}^{\infty }\Phi_{j}\left(x\right){\ddot{q}}_{j}\left(t\right)$. In most cases, a small number j=N of low-frequency eigenvalue pairs is required for acceptable accuracy in modelling the structure in question for external vibrations. Thus, for Ν>1, the extraction of the eigenvectors using the aforementioned architecture and their verification through comparison with the Principal Component Analysis (PCA) requires the adoption of a different *cost* function, separate from the standard one [22]. However, if the transient records are produced using only one eigenvector, then the use of the standard *cost* function yields the same eigenvectors as those derived from the use of the PCA. Note that for a continuous mass representation of the pylon, we use eigenfunctions, while in the case of experimental measurements, these eigenfunctions actually become eigenvectors as they are measured at discrete points along the height of the pylon.

In reference to Figure 2, the elasticity modulus and the mass density of the pylon were experimentally determined, followed by the quantification of equivalent springs to model the loss of fixity at the pylon’s base. The cases pertaining to the first two pylon scenarios can be handled within the linear material response range. References [23,24,25] give many details on the placement of the mass at the top of the pylon and the insertion of base springs, plus their effect on the pylon’s dynamic response. The ratio of the top mass to the total pylon mass R=0.17 is non-negligible, i.e., it is sufficient to alter the eigenvalues of the stand-alone pylon.

## 3. Mechanical Model for the Pylon

The governing equations of motion for the tapered, cantilevered beam with distributed mass derive from the equilibrium diagram of a differential segment dx under distributed longitudinal p(x,t) and transverse f(x,t) load, see Figure 3:
(1)$$\frac{\partial }{\partial x}\left(EA(x)\frac{\partial u(x,t)}{\partial x}\right)-m(x)\frac{\partial^{2}u(x,t)}{{\partial t}^{2}}=p(x,t)$$
(2)$$\frac{\partial^{2}}{{\partial x}^{2}}\left(EI(x)\frac{\partial^{2}w(x,t)}{{\partial x}^{2}}\right)+m(x)\frac{\partial^{2}w(x,t)}{{\partial t}^{2}}=f(x,t)$$

In the above, u(x,t) and w(x,t) the longitudinal and transverse displacements, respectively, while EA(x) and EI(x) are the axial and bending stiffness of the pylon. Furthermore, m(x) is the mass per unit length, and L=b−a is the height. The axial force, bending moment, and shear force, respectively, are:(3)$$N=EA(x)(\partial u/\partial x),\;M=-EI\left(x\right)\left({\partial w}^{2}/{\partial x}^{2}\right),\;Q=-EI\left(x\right)\left({\partial w}^{3}/{\partial x}^{3}\right)$$

Finally, initial conditions are assumed to be zero, while the boundary conditions for a fixed-base pylon are:(4)$$M\left(a,t\right)=Q\left(a,t\right)=N\left(a,t\right)=0$$
(5)$$w\left(b,t\right)=\partial w(b,t)/\partial x=u\left(b,t\right)=0$$

### 3.1. Axial Vibrations

Given a harmonic base motion in the vertical direction of $u_{g}\left(t\right)=u_{go}e^{i\Omega t}$ with amplitude $u_{go}$ and excitation frequency Ω, the axial force is $p\left(x,t\right)=-m\left(x\right){\ddot{u}}_{g}\left(t\right)=m\left(x\right)\Omega^{2}u_{go}e^{i\Omega t}$ and the corresponding axial displacement is $u(x,t)=u_{o}(x)e^{i\Omega t}$.

We define here the reference case of a uniform pylon with a cross-section radius equal to that at the top, i.e., r=r(a). The equation of motion is now
(6)$$u_{o}^{\prime \prime }+{\hat{z}}^{2}u_{o}=-{\hat{z}}^{2}u_{go},\hat{z}=C_{P}^{-1}\sqrt{1/\left(1+i\delta \right)}\Omega$$
where $C_{P}=\sqrt{\hat{E}/\rho }$ is the pressure wave velocity, ρ is the mass density, and δ is the dimensionless damping factor for viscoelastic material behavior. Next, we define the ratio of top-to-bottom pylon displacements ${U_{0}=u}_{0}(x=a)/u_{g0}$, which is the complex-valued transfer function with $\theta_{0}$ as the phase angle:(7)$$U_{0}\left(\Omega \right)=sec(\hat{z}L)-1=\left|U_{0}(\Omega )\right|e^{i\theta_{0}(\Omega )}$$

Consider now the non-uniform cantilevered pylon with a variable cross-section area of $A\left(x\right)=A_{o}{\left(x/\alpha \right)}^{2},\;a\leq x\leq b$. The equation of dynamic equilibrium for harmonic base motion is:(8)$$x^{2}{u_{o}}^{\prime \prime }+2x{u_{o}}^{\prime }+{\hat{z}}^{2}x^{2}u_{o}=-{\hat{z}}^{2}x^{2}x_{go}$$

The above is a Bessel equation whose solution is u0x=x−12(c1 J1/2z^x+c2J−1/2z^x)−ugo, where $J_{n}$ are Bessel functions of the first kind and ½ order. Imposition of the fixed base boundary conditions yields the following normalized transfer function:(9)$$U_{0}\left(\Omega \right)=-1+\frac{b}{a}\frac{\{\hat{z}a\cos\left(\hat{z}a\right)-\sin(\hat{z}a)\}\cos\left(\hat{z}a\right)-\left\{-\hat{z}a\sin\left(\hat{z}a\right)-\cos\left(\hat{z}a\right)\right\}\sin\left(\hat{z}a\right)}{\{\hat{z}a\cos\left(\hat{z}a\right)-\sin(\hat{z}a)\}\cos\left(\hat{z}b\right)-\left\{-\hat{z}a\sin\left(\hat{z}a\right)-\cos\left(\hat{z}a\right)\right\}\sin\left(\hat{z}b\right)}$$

### 3.2. Flexural Vibrations

A forcing function of amplitude with amplitude $w_{go}$ is now applied to the pylon in the horizontal direction, i.e., $f\left(x,t\right)=-m\left(x\right){\ddot{x}}_{g}\left(t\right)=m\left(x\right)\Omega^{2}w_{go}e^{i\Omega t}$,. For the reference case of the uniform pylon, the governing equation is a fourth-order ordinary differential equation:(10)$$w_{o}^{\prime \prime \prime \prime }-{\hat{z}}^{4}w_{o}={\hat{z}}^{4}w_{go},{\hat{z}}^{4}=\rho A\Omega^{2}/ΕI(1+i\delta )$$

The transverse displacement response is $w(x,t)=w_{o}\left(x\right)e^{i\Omega t}$, with the amplitude $w_{o}\left(x\right)=\left|w_{o}\left(x\right)\right|e^{i\theta }$.

Details regarding the solution procedure can be found in [26,27], and the normalized transfer function for the transverse vibrations is defined as ${W_{0}=w}_{0}\left(x=L\right)/w_{go}$. More specifically, we have
(11)$$W_{0}\left(\Omega \right)=\left(c_{1}\sinh\left(\hat{z}L\right)+c_{2}\cosh\left(\hat{z}L\right)+c_{3}\sin\left(\hat{z}L\right)+c_{4}\cos\left(\hat{z}L\right)\right)-1$$
with the constants appearing above defined as
(12)$$c_{1}=\frac{-cosh(\hat{z}L)\sin\left(\hat{z}L\right)-sinh(\hat{z}L)\cos\left(\hat{z}L\right)}{2+2cosh(\hat{z}L)\cos\left(\hat{z}L\right)}\;c_{2}=\frac{sinh(\hat{z}L)\sin\left(\hat{z}L\right)+cosh(\hat{z}L)\cos\left(\hat{z}L\right)+1}{2+2cosh(\hat{z}L)\cos\left(\hat{z}L\right)}\;c_{3}=\frac{cosh(\hat{z}L)\sin\left(\hat{z}L\right)+sinh(\hat{z}L)\cos\left(\hat{z}L\right)}{2+2cosh\hat{z}L\cos\hat{z}L}\;c_{4}=\frac{1-sinh(\hat{z}L)\sin\left(\hat{z}L\right)+cosh(\hat{z}L)\cos\left(\hat{z}L\right)}{2+2cosh(\hat{z}L)\cos\left(\hat{z}L\right)}$$

Continuing with the tapered pylon that has both a variable cross-section area and moment of inertia $Ι\left(x\right)={I_{o}(x/a)}^{6}$ and introducing the dimensionless spatial variable ξ=x/a, yields the equation of motion as:(13)$$\xi^{4}w_{o}^{\prime \prime \prime \prime }\left(\xi \right)+12\xi^{3}w_{o}^{\prime \prime \prime }\left(\xi \right)+30\xi^{2}+w_{o}^{\prime \prime }\left(\xi \right)-{\hat{z}}^{4}w_{o}\left(\xi \right)={\hat{z}}^{4}w_{go}$$
with parameter ${\hat{z}}^{4}={\rho A_{o}\Omega^{2}a}^{4}/\hat{Ε}{Ι}_{ο}$. The solution to this fourth-order Euler equation is:(14)woξ=c1A1ξ+c2A2ξ+c3A3ξ+c4A4ξ−wgo=c1ξ−32+k^+172+c2ξ−32−k^+172+c3ξ−32 sin⁡k^−172ln⁡ξ+c4ξ−32 cos ⁡k^−172ln⁡ξ−wgo
where $\hat{k}=4\;\sqrt{{\hat{z}}^{4}+4}$ is a parameter. Finally, imposition of the boundary conditions gives a system of algebraic equations for the constants appearing in the above solution:(15)$$\left[\begin{array}{cc} \begin{array}{cc} A_{1}(b/a) & A_{2}(b/a)\\ A_{1}^{\prime }(b/a) & A_{2}^{\prime }(b/a) \end{array} & \begin{array}{cc} A_{3}(b/a) & A_{4}(b/a)\\ A_{3}^{\prime }(b/a) & A_{4}^{\prime }(b/a) \end{array}\\ \begin{array}{cc} A_{1}^{\prime \prime }\left(1\right) & A_{2}^{\prime \prime }(1)\\ A_{1}^{\prime \prime \prime }\left(1\right) & A_{1}^{\prime \prime \prime }(1) \end{array} & \begin{array}{cc} A_{3}^{\prime \prime }(1) & A_{4}^{\prime \prime }(1)\\ A_{3}^{\prime \prime \prime }(1) & A_{4}^{\prime \prime \prime }(1) \end{array} \end{array}\right]\cdot \left\{\begin{array}{c} c_{1}\\ \begin{array}{c} c_{2}\\ c_{3} \end{array}\\ c_{4} \end{array}\right\}=\left\{\begin{array}{c} w_{go}\\ \begin{array}{c} 0\\ 0 \end{array}\\ 0 \end{array}\right\}$$

The solution at the top of the pylon, where ξ=1.0, simplifies because $A_{1}\left(1\right)=A_{2}\left(1\right)=A_{4}\left(1\right)=1,\;{\;A}_{3}\left(1\right)=0$. Thus, the transfer function simply becomes:(16)$$W_{0}(\Omega )=(1/x_{go})\cdot (c_{1}+c_{2}+c_{4})-1$$

## 4. Industrial Pylon Testing

A comprehensive experimental project was conducted to study the mechanical behavior of industrial pylons used for lighting highways [26]. Fatigue considerations, dynamic response, cyclic loading, and base conditions are all examined for structural health monitoring (SHM) purposes, including the development of CNN, which is reported here.

Two 2.48m tall metallic pylons with a variable, ring-type cross-section provided by the manufacturer were tested in the AUTh Laboratory for Experimental Mechanics under fixed-base conditions, with one damaged after cyclic testing and the other remaining intact, as shown in Figure 4. Next, Figure 5 is a schematic of the testing setup, while Table 1 lists the mechanical properties and the geometry of the tested pylons. The effect of an external force, as applied to the attached top mass on the pylon’s vibrations is described in Appendix A.



E


GPa


ρ


$tn/m^{3}$


r(a)


mm


r(b)


mm


d


mm


L


m


$K_{X}$


∞


KN/m


$K_{Y}$


∞


KN/m


$K_{\Theta }$


KNm/rad



An important design consideration was fatigue, as these pylons are exposed year-round to wind pressure and rain. Corrosion resistance is addressed by a zinc coating of the external surface, which provides negligible additional strength to the pylon. Fatigue testing showed that after 80,000 cycles of low frequency and low amplitude loading, the pylon developed hairline cracks at its base, whose presence cannot be easily detected by dynamic tests. These cracks, however, act as stress concentrators with the potential to compromise the structural integrity of the pylon. More specifically, impact hammer tests, which followed before and after the cyclic testing, showed no measurable change in the dynamic properties of the cantilevered pylon. Next, a digital image correlation (DIC) system was implemented on an intact pylon as a means of tracing the development of the kinematic field from a series of time frames depicting the displacement evolution over the front surface of the pylon. These DIC studies, conducted for cyclic loads, were in excellent agreement with the conventional method of tracing the displacement vector field at a fixed location on the pylon. Overall, good agreement was achieved between the experimentally obtained results and the analytical/numerical predictions.

Table 2 lists some of the numerically obtained results derived from the use of the analytical models previously developed that pertain to the recovery of the eigenfrequencies of the pylon under the four scenarios discussed earlier on. Also, Figure 6 shows the close agreement between the computed and the experimentally measured first flexural eigenfunction $\varphi_{1}$ of the tapered, cantilevered pylon versus normalized pylon height ξ. In sum, the experiments carried out with the two pylons were used to verify the analytical-numerical models developed for pylon vibrations under intact conditions, base damage and base fixity. Therefore, the analytical models will be used from now on to produce data for the CNN that is being developed in this work.

### Data Generation from Testing under Dynamic Conditions

Dynamic testing, as realized by imparting an impact loading to the pylon with an impact hammer, was the most relevant to CNN development. More specifically, Table 3 gives a description of the wireless acceleration sensors and their accompanying data aggregator, while Figure 7 depicts the placement of six acceleration sensors along the height of the pylon along with the measurements carried out for the four scenarios previously described. Furthermore, Table 4 depicts the drift in the first eigenvalue of the originally intact pylon as the crack develops after the cyclic testing is completed and followed by the impact test. We observe that this drift is negligible and does not allow for crack detection. Because of the low-frequency testing, measurements did not show any contribution from the higher eigenfrequencies to the total dynamic response of the pylon.



±8 g 


$25\;mg/\sqrt{Hz}\;$


20 bit 


256 Hz





$\mathrm{Numerical}\;f_{1}\;\left[Hz\right]$


$\mathrm{Experimental}\;f_{1}\;\left[Hz\right]$



As far as the extraction of the corresponding eigenvectors from the measured response of the pylon is concerned, it was accomplished by the use of *autoencoders* following the architecture outlined in Figure 1.

In training the neural network, the *adam* algorithm was used [30] with a degree of learning a=0.001, while 50 *epochs* were required for stabilization of the *loss* function. All time histories were normalized such that their maximum value over the entire time length was equal to unity. Following training, the CNN was used to extract the eigenfunctions φ(x) listed in Table 5, as well as the generalized coordinate q(t) for each case, see Figure 8.

## 5. CNN Realization

Following the classification of damage in four categories (scenarios), it remains difficult to identify each of them from measurements involving the first eigenvalue–vector pair, which is the only possibility given the low-frequency vibration environment realized in the testing. An earlier neural network implementation by the authors [27] involved a statistical analysis using Spearman correlation coefficients. This method quantifies the influence of varying soil stiffness, mass ratios, and tapering on the pylon’s response while excluding considerations of interior damage. This latte type of nonlinearity prompted the development of a special-purpose CNN capable of distinguishing among four damage categories. A necessary first step in this process is the preparation of the experimentally obtained data. More specifically, the data used come from an acceleration sensor **a805** placed at the top of the pylon, see Figure 7. The time histories are converted into spectrograms [31], which are used as input to the CNN.

For each of the four scenarios, we use the first 200 s of the time histories to set up both a training and a validation set. Following the development of CNN, we use the remaining 25 s, 35 s, and 257 s record snippets to create test sets that would check the CNN’s performance. Note that the record duration for the first scenario is 225 s, that of the second scenario is 235 s, and finally, that of the last two scenarios is 457 s.

### 5.1. Training and Validation Data Sets

As mentioned, training sets were developed using the first 200 s of each experimentally obtained time history record. Specifically, a total of four sets containing 100 time histories were prepared (see Figure 9), with each set corresponding to a damage scenario. These were used for the CNN training and validation, while at the same time, 281 time histories were reserved for testing the performance of the CNN. These last sets were broken down as follows: (a) 11 time histories for the undamaged state, (b) 16 time histories for the undamaged pylon with an attached top mass, (c) 127 time histories for the damaged pylon with a base crack and with recordings available in the N–S direction only, and (d) similarly with 127 time histories coming from the E–W direction.

Next is the development of a total of 100 spectrograms of 4s duration, with each spectrogram overlapping the next one by 2 s. In their development, a Hanning window [31] with an element size of 256 and an element overlap of 250 was used. Figure 10 depicts four typical spectrograms, each pertaining to one of the four damage scenarios. Specifically, a comparison of cases (a) and (b) indicates that in the latter case, the addition of a top mass with a mass ratio of R=0.17 renders the pylon more flexible. This is manifested by the fact that the indigo-colored horizontal line in Figure 10b, which indicates the time variation of the first eigenfrequency, is lower than the corresponding one in Figure 10a, attesting to a more flexible structural system. Also, the remaining faint indigo-colored horizontal lines in the spectrograms correspond to some parasitic eigenfrequencies brought about by some amount of flexibility in the base connection of the pylon during the experiments. Next, the blue-colored vertical lines in the spectrograms pertain to the instant when an impact was delivered to the pylon through the use of the impact hammer, which excites a wide range of frequencies, and those are registered in the response. Finally, the green-colored areas correspond to a free-vibration environment. In this case, the last two spectrograms have identical lines, as they both derive from the same impact.

### 5.2. CNN Architecture

The CNN architecture, shown in Figure 11, consists of three convolution layers. The first layer comprises 32 filters (or kernels) size 6 × 6, the second layer comprises 64 filters size 3 × 3, while the third one has 128 filters, also of size 3 × 3 each. Past the end of each filter is a *max pooling* operator, whose purpose is to reduce the size of *feature maps* without compromising the integrity of the information contained there. Next, a *dense layer* comprising 128 neurons is placed to correspond to the *feature maps* with an optimal vector size of *4* × *1*, where 4 is the number of classes. Finally, the *dropout* technique is used to handle problems with over-fitting. More specifically, during the training period, this technique ignores a cluster of neurons, forcing the network to seek a plurality of characteristics instead of the ones currently available.

The *cost* function used in the CNN development is the *sparse categorical cross-entropy*, while the computation of weights materializes through the use of the *adam* algorithm with a learning rate of *a = 0.001*. The total development session of the CNN lasted 100 *epochs*, with 80% of the 400 spectrograms used for training and the remaining 20% for validating; see Figure 12 for details. Following training, the *confusion matrix* shown in Table 6 lists the results of the validation session, indicating a high level of accuracy in the results. Note that the CNN validation was accomplished with spectrograms that were not used in the training session.

The CNN was used to evaluate damage from the remaining spectrograms that were reserved for a true test of CNN capabilities, and the results are shown in the confusion matrix in Table 7 and Table 8. Since we have leakage because of the presence of non-diagonal terms in the *confusion matrix*, a careful assessment of the CNN performance is warranted. We first start with each damage scenario (or class) and compute the following variables: *True Positives* (TP), *False Negatives* (FN), and *False Positives* (FP). We observe that class (a) receives the lowest precision. Specifically, even if the input is unbalanced, the *confusion matrix* indicates that if the CNN performs 100 evaluations and classifies them as belonging to scenario (a), then 58.8% (10/17) are correct, and 41.2% (7/17) are misclassified as class (d) meaning that there is crack damage in direction *N–S*, but this damage does not show along direction *W–E* and the pylon appears to be undamaged. This has important implications, meaning that when monitoring pylons of circular cylindrical geometry, sensors must record data along two perpendicular directions. Notice that cracks are linear elements, and when section bending occurs, the crack might be inactive if it is subjected to compression, which results in closing the two crack surfaces. This will mask the presence of the crack unless measurements are taken in the directions where the crack opens up. Should this happen, the CNN interpretation of the measurements will be reliable, and a high percentage will appear in both the precision and recall indices, which will provide a high degree of confidence for the practicing engineer’s purposes.

## 6. Conclusions and Discussion

In SHM, detection of damage remains difficult (unless it is already visible) despite the generation of extensive data, which has to be processed nowadays by artificial intelligence (AI) techniques. Specifically, in this work, a CNN was developed to determine damage in an industrial pylon by juxtaposing two basic states, the undamaged and the damaged ones. Dynamic testing was conducted by using an impact hammer, while time recordings were available from a number of wireless acceleration sensors. Each state further contained two sub-cases, the former having the stand-alone case and the pylon with an attached heavy mass, while the latter dealt with cracking at the base of the pylon following extensive fatigue testing, but with recordings available along the crack and perpendicular to the crack directions.

The basic conclusion of this experimental-numerical work is that damage is difficult to detect at its early stages, and sometimes, the damaged state cannot be identified, and CNN confuses it with undamaged states. This requires measurements taken in the vicinity of the suspected damage area and in the principal directions of the mechanical response. This is because damage may be detected in the transverse vibrations only, or perhaps in the axial vibrations or even in the torsional response.

It was not possible to compare results with similar situations where the response of the pylon is nonlinear and requires data-driven models, despite the fact that the literature on pylons pertaining to wind turbine design is vast, with the focus being on passive and active control mechanisms. This way, the continuous operation of the wind turbine is better secured. For less important categories of pylons, e.g., lighting posts, SHM consists of visual inspections. However, given the proliferation of such categories of pylons, for which vast numbers are produced, it is important to have SHM strategies validated through benchmark tests in place.

In closing, the pylon’s vibrations came about from random impact hammer hits along the pylon’s surface and at different time instances. It is interesting to speculate if the confusion matrices developed for these cases would exhibit similar accuracy with the one presented herein if the cause of vibration was either base motion (e.g., earthquake or machine vibrations) or top motion (e.g., the presence of a rotor).

## Figures and Tables

**Figure 1 sensors-24-06255-f001:**
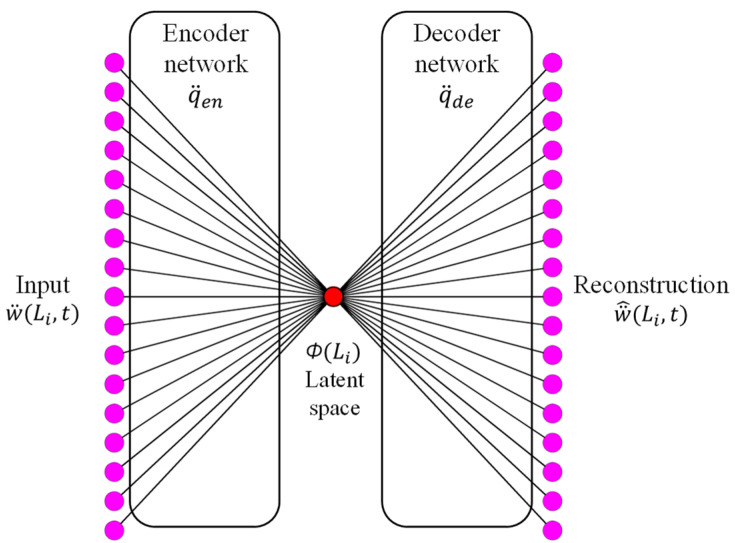
Typical *autoencoder* architecture.

**Figure 2 sensors-24-06255-f002:**
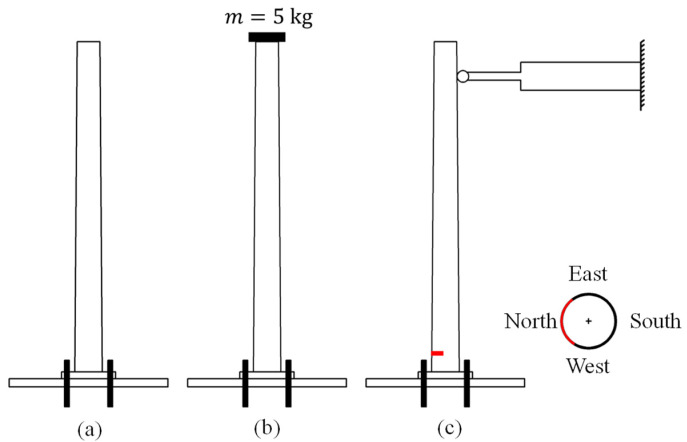
Flexible pylon configurations: (**a**) Intact condition; (**b**) placement of a 5 kg mass at the top; (**c**) crack development at the base during a fatigue test involving a large number of cycles induced by an actuator at the top.

**Figure 3 sensors-24-06255-f003:**
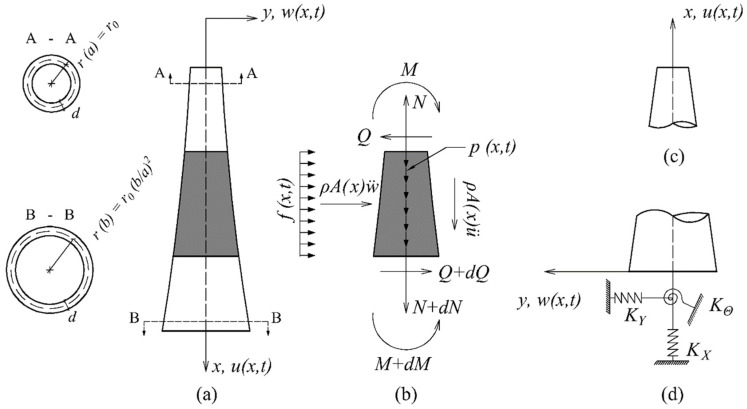
Cantilevered pylon: (**a**) cross section and length geometry; (**b**) free-body diagram; (**c**) top segment detail; (**d**) pylon base details.

**Figure 4 sensors-24-06255-f004:**
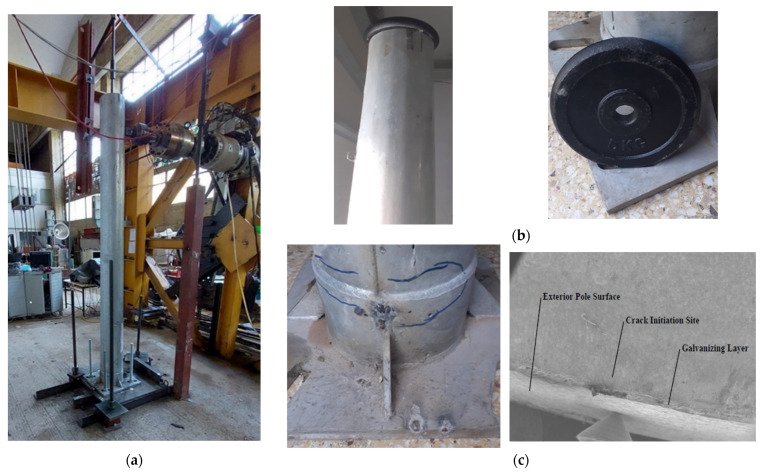
(**a**) Cantilevered pylon under cyclic loading; (**b**) details of the top mass; (**c**) details of cracking formation at the pylon’s base [28].

**Figure 5 sensors-24-06255-f005:**
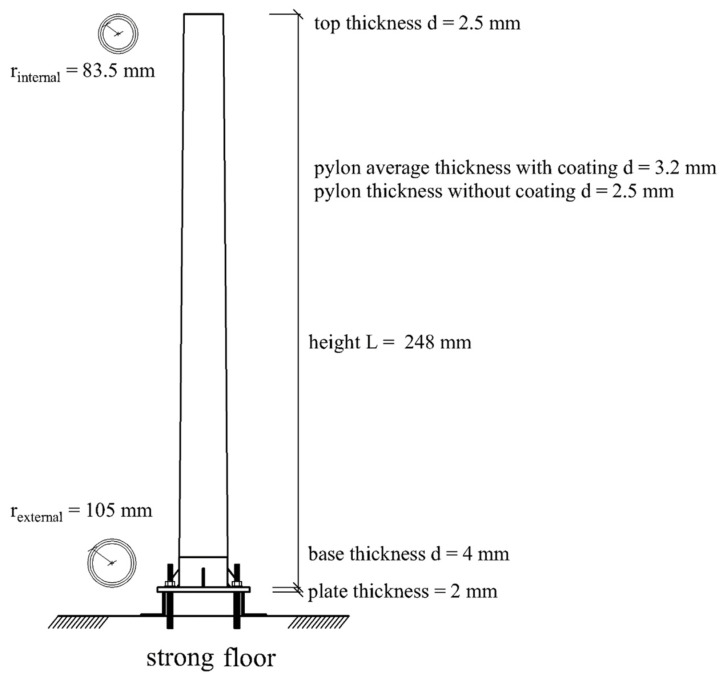
Cantilevered pylon mechanical model.

**Figure 6 sensors-24-06255-f006:**
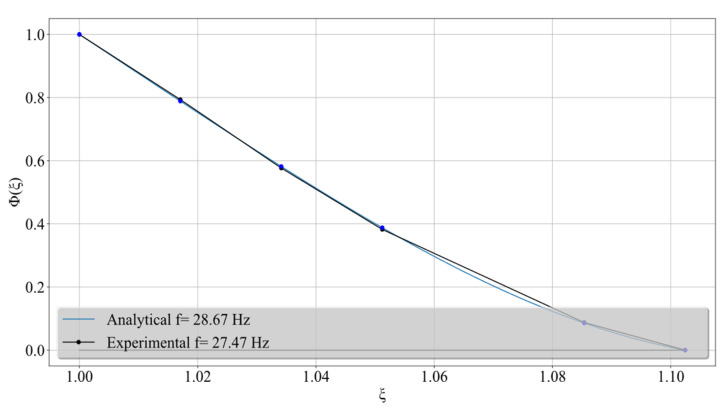
Comparison between the analytically computed and experimentally measured first flexural eigenfunction $\varphi_{1}$ of the tapered pylon versus normalized pylon height ξ.

**Figure 7 sensors-24-06255-f007:**
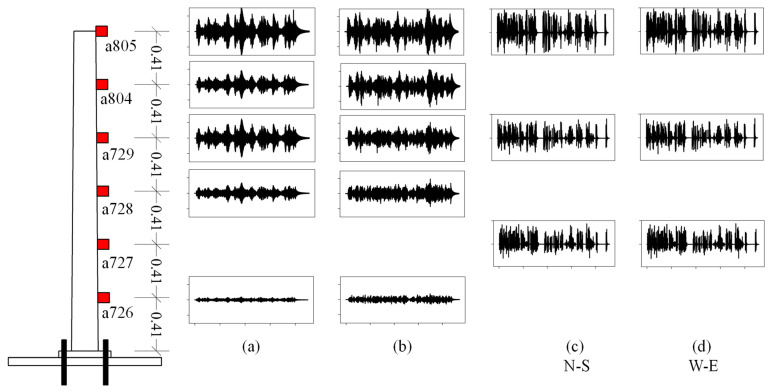
Pylon instrumentation (left) and typical acceleration time histories recorded by the sensors for cases: (**a**) Intact pylon, 225 s duration; (**b**) Pylon with attached mass at the top, 225 s duration; (**c**) Pylon with a base crack and measurements in the N–S direction, 457 s duration; (**d**) Pylon with a base crack and measurements in the E–W direction, 457 s duration.

**Figure 8 sensors-24-06255-f008:**
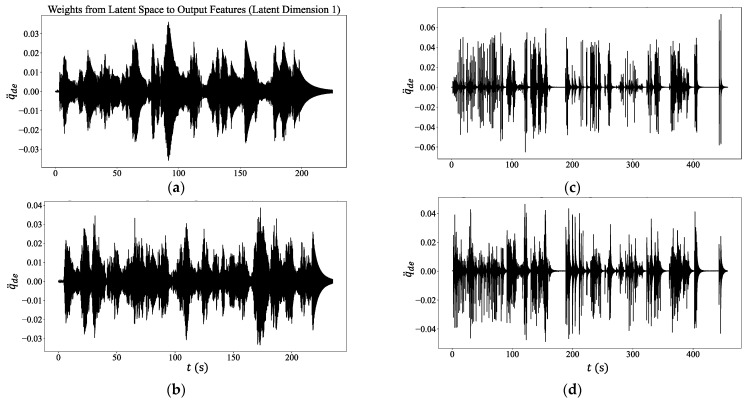
The pylon’s generalized coordinates as derived from the weight functions of the Autoencoder: From top downwards are cases (**a**) intact pylon, (**b**) intact pylon with a top mass, (**c**) pylon with a base crack and measurements in the N–S direction, (**d**) pylon with a base crack and measurements in the W–E direction.

**Figure 9 sensors-24-06255-f009:**
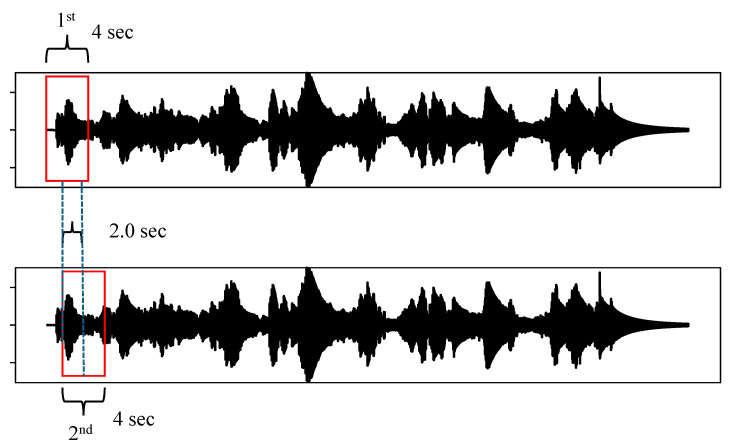
The first two time histories out of a bin of 400 time histories, recorded by sensor **a805,** that are generated as input for the development of spectrograms.

**Figure 10 sensors-24-06255-f010:**
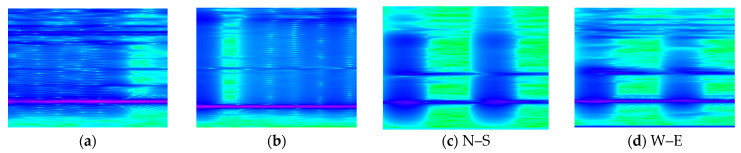
Representative spectrograms for pylon scenarios: (**a**) Intact, (**b**) intact pylon with a top mass, (**c**) pylon with a base crack and N–S recording, and (**d**) pylon with a base crack and W–E recording.

**Figure 11 sensors-24-06255-f011:**
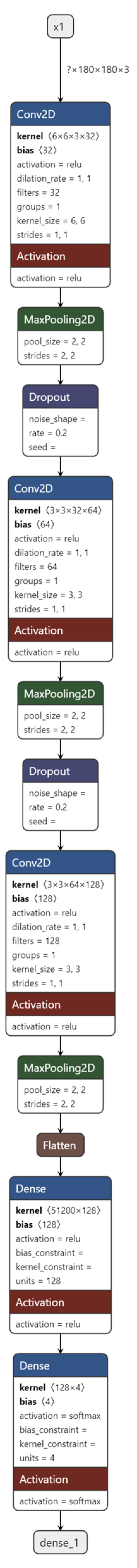
CNN architecture: Input, neuron connections, and output [32]. Note: The question mark (?) at the top was set equal to one in our development.

**Figure 12 sensors-24-06255-f012:**
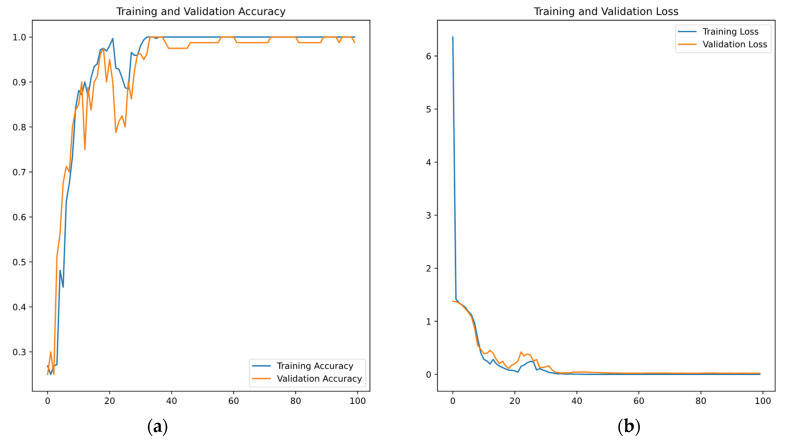
CNN development: (**a**) Increase in accuracy and (**b**) decrease in the value of the *loss* function as a function of the *epochs*.

**Table 1 sensors-24-06255-t001:** Mechanical properties and geometry of the variable cross-section pylon [26].

Property/Dimension	Symbol	Value	Units
Modulus of elasticity	E	230	GPa
Mass density	ρ	7.88	$tn/m^{3}$
Cross-section mean radius at the top	r(a)	84.8	mm
Cross-section mean radius at the base	r(b)	103.0	mm
Cross-section effective thickness	d	2.5	mm
Pylon height	L	2.48	m
Equivalent base springs	$K_{X}$	∞	KN/m
	$K_{Y}$	∞	KN/m
	$K_{\Theta }$	4621	KNm/rad

**Table 2 sensors-24-06255-t002:** Sequence of eigenfrequencies $f_{n}\left(Hz\right),n=1,2,3$ of the industrial pylon.

Eigenfrequencies:	*n* = 1	*n* = 2	*n* = 3
Axial vibrations, homogeneous pylon	544.57	1633.7	2722.8
Axial vibrations, tapered pylon	566.28	1641.2	2727.3
Flexural vibrations, homogeneous pylon	32.63	204.46	572.50
Flexural vibrations, tapered pylon	36.57	211.27	577.70
Flexural vibrations, tapered pylon with rotational base spring	28.67	177.54	505.50

**Table 3 sensors-24-06255-t003:** Wireless sensor network specifications [29].

Wireless Triaxial Accelerometer Node G-Link-200		Wireless Sensor Data Aggregator WSDA-2000	
Measurement range	±8 g	Radiofrequency transceiver carrier	License-free 2.405 to 2.480 GHz With 16 channels
Noise density	$25\;mg/\sqrt{Hz}$		
Resolution	20 bit		
Sampling ratio	256 Hz		

**Table 4 sensors-24-06255-t004:** Measured and computed first eigenfrequencies $f_{1}$ of the cantilevered pylon from the impact tests for all four scenarios.

	**(a)**	**(b)**	**(c)**	**(d)**
$\mathrm{Numerical}\;f_{1}\;\left[Hz\right]$	28.67	22.07	-	-
$\mathrm{Experimental}\;f_{1}\;\left[Hz\right]$	27.47	21.86	27.46	27.60

**Table 5 sensors-24-06255-t005:** Differences in the values of the eigenvectors as determined by the use of the *autoencoder* (label Ι) and principal component analysis, PCA (label II). All eigenvectors are derived from the corresponding sensors are normalized with a unit value at the pylon’s top.

	Eigenvectors (a)		Eigenvectors (b)		Eigenvectors (c)		Eigenvectors (d)	
	I	II	I	II	I	II	I	II
**a805**	1.00	1.00	1.00	1.00	1.00	1.00	1.00	1.00
**a804**	0.79414	0.793970	0.787044	0.789250	-	-	-	-
**a729**	0.58039	0.577731	0.570007	0.572198	0.558527	0.5715071	0.5697331	0.554934
**a728**	0.38630	0.38368	0.374974	0.379083	-	-	-	-
**a727**	-	-	-		0.23591	0.239123	0.235503	0.237070
**a726**	0.087738	0.08767	0.084593	0.085166	-	-	-	-

**Table 6 sensors-24-06255-t006:** The *confusion matrix* computed from the validating data set.

Predicted	(a)	20	0	0	0
	(b)	0	20	0	0
	(c) N–S	0	0	20	1
	(d) W–E	0	0	0	19
		(a)	(b)	(c) N–S	(d) W–E
		**True**			

**Table 7 sensors-24-06255-t007:** The *confusion matrix* computed from the testing data set.

Predicted	(a)	10	0	0	7
	(b)	0	14	0	0
	(c) N–S	1	2	124	5
	(d) W–E	0	0	2	115
		(a)	(b)	(c) N–S	(d) W–E
		True			

**Table 8 sensors-24-06255-t008:** The confusion matrix computed from the metrics that quantify CNN performance.

			$Precision=\frac{TP}{TP+FP}$	$Recall=\frac{TP}{TP+FN}$
(a)	TP	10	0.588	0.909
	FN	1		
	FP	7		
(b)	TP	14	1.00	0.875
	FN	2		
	FP	0		
(c) N–S	TP	124	0.939	0.984
	FN	2		
	FP	8		
(d) W–E	TP	115	0.983	0.906
	FN	12		
	FP	2		

## Data Availability

Data is available upon request.

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
