# Peer review of "Convolution Neural Network Development for Identifying Damage in Vibrating Pylons with Mass Attachments"

_sensors, 2024, doi:10.3390/s24196255_

Round 1

Reviewer 1 Report

Comments and Suggestions for Authors

In this article, the authors introduce a Convolution Neural Network (CNN)to detect damage in pylons from measurement of their vibratory response. The input of the neural network is the spectrograms obtained from the four types of data collected by the sensor. Through training with different spectrograms features, this approach is proposed to aim at predicting the state of different categories. The presentation of the results is also acceptable. However, still some crucial points need to be addresses before the final decision. The details are listed below : 

(1) The literature review is not enough, and the research status of the field is not enough.  

(2) In this paper, data-driven methods are used, but the comparison between traditional methods and data-driven methods is lacked. So why do the authors use a data-driven approach? 

(3) The lines of figures in Figure 3, Figure 5 and Figure 11 are blurred, please provide clear pictures. 

(4) I suggest that the network structure information of convolutional neural networks should be provided in a table to show clearly. 

(5) In this paper, no comparative test was conducted to discuss the advanced nature of the proposed method, please provide relevant comparative tests.

Comments on the Quality of English Language

The quality of English language seems fine, and some minor errors should be modified.

Reviewer 2 Report

Comments and Suggestions for Authors

The authors conducted a series of full-scale tests to investigate the effectiveness of a convolution neural network (CNN) developed in this work for detecting damage in pylons based on measurements of their vibratory response. Upon reviewing the manuscript, several critical observations were made regarding its content, structure, and clarity.

  1. The manuscript would benefit from a clearer articulation of the importance of damage recognition in the introduction. It is crucial that the research questions driving the study emphasize the significance of the research within the broader field. Additionally, incorporating more contextual background would strengthen the relevance and necessity of the study.

  2. Although the study employs rigorous methods, more detailed information is needed to describe the experimental design and analytical techniques. This information is essential for readers to fully comprehend and potentially replicate the study. The integration of different methodologies, such as industrial pylon testing, supports a comprehensive analysis that accounts for uncertainties. However, the authors should also discuss the limitations of these methodologies and any assumptions made.

  3. The clear presentation of the results is commendable, but the authors are encouraged to include more graphical representations of data where feasible. Visual aids can significantly enhance readers' interpretation and understanding of the findings. Incorporating comparative studies within the results section adds substantial value to the analysis, providing insights into different conditional factors. For instance, including images of hairline cracks observed during industrial pylon testing would be informative.

  4. The discussion section effectively relates the results back to the research questions and highlights their implications, which is a vital component of any academic study. Nevertheless, the authors should provide a deeper analysis by comparing their findings on the digital model for bridge tower damage identification more extensively with existing literature. Furthermore, discussing the conclusions that can be drawn and reflecting on how these results contribute to existing knowledge or practice within the field would strengthen the impact of the study.

Round 2

Reviewer 1 Report

Comments and Suggestions for Authors

The authors have addressed all the concerns, and I have no more comments.